# Unusual Post-Translational Modifications in the Biosynthesis of Lasso Peptides

**DOI:** 10.3390/ijms23137231

**Published:** 2022-06-29

**Authors:** Yuwei Duan, Weijing Niu, Linlin Pang, Xiaoying Bian, Youming Zhang, Guannan Zhong

**Affiliations:** Helmholtz International Lab for Anti-Infectives, Shandong University-Helmholtz Institute of Biotechnology, State Key Laboratory of Microbial Technology, Shandong University, Qingdao 266237, China; yuweiduan@mail.sdu.edu.cn (Y.D.); 201912388@mail.sdu.edu.cn (W.N.); panglinlin991113@163.com (L.P.); bianxiaoying@sdu.edu.cn (X.B.); zhangyouming@sdu.edu.cn (Y.Z.)

**Keywords:** lasso peptides, post-translational modifications, biosynthesis, RiPPs, natural products

## Abstract

Lasso peptides are a subclass of ribosomally synthesized and post-translationally modified peptides (RiPPs) and feature the threaded, lariat knot-like topology. The basic post-translational modifications (PTMs) of lasso peptide contain two steps, including the leader peptide removal of the ribosome-derived linear precursor peptide by an ATP-dependent cysteine protease, and the macrolactam cyclization by an ATP-dependent macrolactam synthetase. Recently, advanced bioinformatic tools combined with genome mining have paved the way to uncover a rapidly growing number of lasso peptides as well as a series of PTMs other than the general class-defining processes. Despite abundant reviews focusing on lasso peptide discoveries, structures, properties, and physiological functionalities, few summaries concerned their unique PTMs. In this review, we summarized all the unique PTMs of lasso peptides uncovered to date, shedding light on the related investigations in the future.

## 1. Introduction

Ribosomally synthesized and post-translationally modified peptides (RiPPs) are a family of natural products with remarkable structural variety and functional diversity due to their extensive post-translational modifications (PTMs) [1,2]. An intriguing member of RiPPs is lasso peptides that consist of an N-terminal macrolactam ring via an isopeptide bond between the α-amine of the first amino acid residue and the carboxylic acid side chain of an aspartate/glutamate located in the 7–9 residues, and a C-terminal tail threads through the macrolactam to form a characteristic lariat topology. The unique knot-like threaded topology endows most lasso peptides with extraordinary stability against heat, proteolysis, and extreme pH conditions, making them distinct from other RiPPs. Diverse physiological functionalities have been reported for lasso peptides, such as antimicrobial, antitumor, antiviral, and receptor antagonistic activities [3,4,5,6,7].

Generally, at least three gene products are involved in the biosynthesis of lasso peptides: a precursor peptide (A), an ATP-dependent cysteine protease (B), and an ATP-dependent macrolactam synthetase (C). The ribosome-derived linear precursor peptide consists of an N-terminal leader region for the recognition by different PTM enzymes and a C-terminal core region that makes up the structural backbone of lasso peptides. The cysteine protease (B) encompasses a ribosomal recognition element (RRE) domain in the N-terminus to recognize and bind the leader peptide, after which the C-terminal protease domain cleaves the leader peptide to release the leaderless core peptide (prefolded core peptide in Figure 1). In many cases, the N-terminal RRE and the C-terminal protease domains are split into two separate open reading frames (ORFs) termed B1 (RRE) and B2 (cysteine protease), respectively. The nascent core peptide is further delivered to macrolactam cyclase C that uses ATP to activate the side carboxyl of Asp or Glu located at 7–9 as an AMP ester (activated core peptide in Figure 1), followed with the freely N-terminal α-amine attacking the AMP-activated carboxyl group to form an isopeptide bond and achieve the mature lasso peptide (Figure 1). It is very likely that the C-terminal tail of the core peptide is prefolded into the threaded configuration prior to the ring’s closure, otherwise the tail would be excluded by the macrolactam owing to the steric hindrance and unable to form the correct lasso topology. Extra D genes encoding ATP-binding cassette transporters (ABC transporters) are not rare in the biosynthetic gene clusters (BGCs) of lasso peptides and are believed to be responsible for the extracellular transport of mature lasso peptides [3,4,5,6,7].

The inexorable progress in genomics, bioinformatics, and chemical analytics greatly facilities lasso peptides discovery during the last decade. In addition to the class-defining modifications as leader peptide excision and core peptide cyclization, a series of unique PTMs including disulfuration, phosphorylation, C-terminal methylation, acetylation, hydroxylation, etc., have been unveiled recently, further increasing the diversity of structures, properties, and complicating the maturation mechanisms. Herein, we compile all the unique PTMs of lasso peptides uncovered up to now, with the emphasis on the biosynthetic mechanisms in detail.

## 2. Disulfuration

Disulfide bonds are rare even among all known RiPPs families, and may play an auxiliary role in maintaining the correct configurations, which is curial for biological activities. To the best of our knowledge, disulfide bonds are only characterized in three classes of RiPPs: glycocins, the post-translationally glycosylated bacteriocins featuring two nested disulfide bonds that stabilize their unique helix–loop–helix structures and sugar moieties on Ser, Thr, or Cys residues [8]; cyclotides, featuring a head-to-tail cyclic peptide backbone with a cystine knot arrangement of three conserved disulfide bonds [9]; and conopeptides, the cone-snail-derived RiPPs containing a high frequency of PTMs involving disulfide bond(s) [10], albeit a few examples in other classes such as lanthipeptides, cyanobactins, sactipeptides, and lasso peptides also contain disulfide(s). Two thiol-disulfide oxidoreductases and a protein-disulfide isomerase (PDI) were reported for the disulfide bond(s) formation in glycocins and cyclotides, respectively [11,12], while the formation of disulfide bond(s) in conopeptides still remains elusive.

The number and position of disulfide linkage are used to categorize lasso peptides into four classes. Class I lasso peptides contain two disulfide bridges that link the tail above and below the macrolactam ring, and one of the composed Cys residues is located at the N-terminal of the core peptide. Class II lasso peptides are devoid of disulfide bridges, only upheld by a steric lock formed from bulky amino acids placed on both sides of the macrolactam ring, representing the largest category of lasso peptides. Class III and class IV lasso peptides only have one disulfide bond, the difference between the two classes is that the disulfide in class III connects a Cys residue on the ring with the other on the tail, whereas class IV with a disulfide bridge resides completely on the tail (Figure 2a) [3,4,5,6,7].

All characterized class I lasso peptides are isolated from *Streptomyces* and exhibit attractive biological activities, five of which, termed MS-271 (siamycin I) [13], specialicin [14], aborycin (RP-71955) [15], humidimycin [16], and sviceucin [17] share almost the same sequences and BGC arrangements. In silico analysis of the BGC of sviceucin (*svi*) revealed that apart from the fundamental genes coding for the precursor peptide (*sviA*), the split B enzymes (*sviB1*/*B2*), the macrolactam synthetase (*sviC*), the transporters (*sviD1-D4*), and regulators (*sviG*/*R1*/*R2*), two genes related to disulfide bond formation (*sviE*/*F*) were also involved in the downstream of transporter genes. In particular, SviE belongs to the DoxX family enzymes whose function is still ambiguous, SviF contains a thioredoxin domain with a conserved CXXC motif and is homologous to thiol-disulfide oxidoreductases (TDORs) [17]. It was speculated that these two proteins may be responsible for the formation of disulfide bonds, and their homologous genes also exist in four other BGCs of class I lasso peptides (Figure 2b) [13,14,15,16,17]. However, sviceucin could still be generated in the *ΔsviF* mutant strain and was only detected in the mycelia instead of present in both culture supernatant and mycelia, which suggested that SviF is dispensable for the disulfide bond formation in sviceucin but plays an essential role in mature peptide transport [17]. Similarly, MS-271, isolated from *Streptomyces* sp. M-271, could also be detected when *mslE* and *mslF* (analogous to *sviE* and *sviF*) were knocked out, respectively, albeit the yields were diminished, and no mercapto intermediates were detected [13]. In addition, the homologous genes of *sviE* and *sviF* are absent in the BGC of arcumycin, another class I lasso peptide isolated from *Streptomyces* sp. NRRL F-5639 very recently (Figure 2b) [18], further implying that the disulfide oxidoreductases are dispensable for disulfide formation.

The mysterious class III lasso peptides comprise very few representatives including BI-32169 [19] and 9401-LP1 [20]. The BGC of BI-32169 has not yet been identified as far as we know, whereas the BGC of 9401-LP1 included no extra genes other than A-D genes (Figure 2c) [20]. Very little is known about the biosynthesis of class III lasso peptides.

The BGC of the class IV lasso peptide felipeptins encodes a predicted flavin-dependent oxidoreductase FilE (Figure 2d). Despite lacking any experimental evidence, this protein was speculated to form the disulfide bridge located in the threaded tail [21]. However, no *filE* homologous genes are founded in the BGCs of the other two reported class IV lasso peptides LP2006 [22] and pandonodin [23]; thus, the exact function of FilE still remains obscure. It is worth noting that no extra homologous genes other than the essential A-D genes are located in the class IV BGCs of LP2006, pandonodin, and felipeptins (Figure 2d). It is conceivable that the disulfuration is a non-enzymatic modification.

Natural stlassin is a class II lasso peptide without any extra PTMs (Figure 2a). Double cysteine mutant variants Val2Cys/Ala11Cys and Val3Cys/Pro12Cys produced two stlassin derivatives with disulfide bonds positioned between the macrolactam ring and the C-terminal loop, creating a type of lasso fold that is outside the traditional four classes [24]. Notably, these derivatives support that disulfide bonds in lasso peptides might form spontaneously, consistent with the findings from class I and IV lasso peptides. Further investigation is required to clarify this controversial mechanism.

## 3. Phosphorylation

Phosphorylation was the earliest characterized tailoring process in lasso peptides. Paeninodin, originated from firmicute strain *Paenibacillus dendritiformis* C454, is a class II lasso peptide with a Ser residue in the C-terminus, of which the BGC encodes an additional putative tailoring kinase (PadeK) (Figure 3a) [25]. Both unphosphorylated paeninodin and phosphorylated paeninodin were detected in the extract of heterologous expression for paeninodin cluster in *Escherichia coli*. Deletion of the kinase gene *padeK* resulted in the production of merely unphosphorylated paeninodin, while restitution of the knocked-out gene by co-expression with another vector-bearing *padeK* led to restoration of the phosphorylated compound, suggesting the direct link between the function of kinase PadeK and the occurrence of the tailoring phosphorylation process on paeninodin. Precursor peptide PadeA instead of the threaded lasso peptide was verified to be the substrate of kinase PadeK, which specifically modified the hydroxyl group of the C-terminal Ser, the extremely conserved site in the precursor sequences from various lasso peptide BGCs featuring a homologous kinase gene, suggesting the modification step prior to the fundamental maturing process catalyzed by B2 and C proteins (Figure 3b) [25]. Owing to the low solubility of PadeK, the homologous kinase ThcoK from another firmicute *Thermobacacillus composti* KWC4 was chosen instead of PadeK to be characterized in vitro. Replacing *padeK* with *thcoK* in the paeninodin heterologous expression system successfully produced the phosphorylated peptide with only minor amounts of unmodified compound, suggesting the feasibility of the hybrid gene cluster. Sequence alignments of lasso peptide-tailoring kinases exposed a conserved His-Lys-Asp-Asp motif. The imperative roles of these four catalytic residues were further demonstrated via site-directed mutations [25].

Subsequent investigation of the kinase ThcoK and another kinase SyanK from proteobacterium *Sphingobium yanoikuyae* ATCC 51230 revealed the attractive polyphosphorylation on their precursor peptides ThcoA and SyanA, despite their low substrate identity except the conserved C-terminal Ser residue [26]. These kinases not only phosphorylated the native precursor peptides but could also modify the phosphate group by stepwise transfer of the second, the third, and even the fourth phosphate group at most (Figure 3c). Tandem mass spectrometry of the lasso peptides confirmed their polyphosphorylated state and the polyphosphorylated position of the C-terminal Ser residue. The degree of modification depends on the donor of the phosphate group (ATP or GTP) along with the sequence of the precursor peptides [26]. The yields are rather low for both monophosphorylation and polyphosphorylation, indicating the possibility that the phosphorylation could be facilitated by the dual functional B1 proteins (*vide infra*).

In addition to the putative kinase gene (*psmK*), an annotated nucleotidyltransferase gene (*psmN*) was identified as well in the BGC of pseudomycoidin from *Bacillus pseudomycoides* DSM 12442. Heterologous expression of the *psm* gene cluster produced both mono- or polyphosphorylated and glycosylated lasso peptides (Figure 3d) and knocking out the related genes validated that PsmK is responsible for the (poly)phosphorylation of the C-terminal Ser residue and PsmN for installing one or two hexose groups on the nascent phosphorylated Ser residue. These results suggested that PsmK is indeed a kinase and PsmN might act as a novel glycosyltransferase homologous to nucleotidyltransferase, appending hexose groups on the phosphorylated peptide. Alternatively, the probability that PsmN is actually a nucleotidyltransferase and the glycosylation is installed after nucleotidylation via a glycosyltransferase present in the *E. coli* host could not be excluded at present [27].

## 4. Methylation

Methylation is a versatile modification in the biosynthesis of various natural products. Lassomycin discovered from *Lentzea kentuckyensis* sp. is an absorbing lasso peptide that exhibits outstanding activities against a variety of *Mycobacterium tuberculosis* strains with minimum inhibitory concentration (MIC) values of 0.8–3 μg/mL and is inactive against symbionts of the human microbiota [28]. Although the initial structure elucidation indicated that lassomycin adopted an unthreaded structure [28], the subsequent chemical synthesis of this peptide showed that the reported structure was incorrect and a characteristic threaded conformation was essential for its anti-tuberculosis (TB) activity [29,30]. In addition, lassomycin features a unique methyl ester in the C-terminal carboxyl group, and the putative *O*-methyltransferase, LasF, from its BGC was considered to be responsible for the C-terminal methylation (Figure 4a) [28].

To understand the biosynthetic pathway of the methylated lasso peptide, two lassomycin-like BGCs from *Sanguibacter keddieii* DSM 10542 (*sake*) and *Streptomyces* sp. Amel2xC10 (*stsp*) containing predicted *O*-methyltransferase genes (*sakeM* and *stspM*) were identified by genome mining (Figure 4a,b). StspM is a novel methyltransferase that specially modified the C-terminal carboxyl of the precursor peptide with a broad substrate specificity, leading the formation of C-terminal methyl carboxylate to precede the maturation process catalyzed by RRE (B1 protein), peptidase (B2 protein), and macrolactam cyclase (C protein) in the lasso peptide biosynthetic pathway (Figure 4c) [31]. Based on homology modeling with mitomycin-7-O-methyltransferase [32] and mutational analysis, His242 and Glu296 were validated as the catalytic center of StspM [31]. Considering the rare *O*-methyltransferases involved in the biosynthesis of lassomycin and two potential lassomycin-like peptides, the biosynthetic mechanism of lassomycin is speculated to follow the same pathway, and the lassomycin-like lasso peptides display identical formability to be candidates for anti-TB drugs.

Interestingly, the structural genes of lassomycin share high homology with the associated genes of lariatin A, another anti-tuberculosis lasso peptide with no tailoring modification [33,34]. Previous studies showed that the C-terminus of lariatin A significantly affected its anti-mycobacterial activity [35,36], which raises the question of whether the C-terminal methylation of lassomycin is equally necessary for its physiological functionality.

## 5. Acetylation

A novel lasso peptide BGC encoded for albusnodin was found in *S. albus* DSM 41398 which includes a putative acetyltransferase gene (*albT*) as well as the canonical genes *albA, albB, and albC* [37]. The only observed heterologous expression product was the threaded, C-terminal Cys truncated albusnodin with an acetyl group attached to the ε-amino group of Lys10 (Figure 5) [37]. Sequence alignments showed that Lys10 was highly conserved among precursor peptides in an array of lasso peptide BGCs that resembled the BGC architecture of albusnodin. Heterologous expression of the albusnodin cluster lacking the acetyltransferase gene *albT* led to no trace of the predicted unacetylated intermediate [37], surmising that the acetylation is vital and occurs in the early stage of albusnodin biosynthesis rather than the last step. Moreover, the BGC of the antitumor lasso peptide ulleungdin also contains an acetyltransferase gene in the downstream of *B2*, yet acetylated ulleungdin was not detected [38]. It seems that this acetyltransferase is unrelated to ulleungdin.

## 6. Hydroxylation

Canucin A and B are 14-mer lasso peptides with identical sequences discovered by eliciting a cryptic BGC (hereafter named *can*) in *S. canus*. The C-terminal Asp14 of canucin A was uniquely *β*-hydroxylated [39]. Bioinformatic analysis showed that the *canE* gene encoding a putative non-heme iron/2-oxoglutarate (Fe/2-OG)-dependent enzyme is involved in this cluster. In vitro characterization of CanE proved that it catalyzed the *β*-hydroxylation on Asp14 in canucin A (Figure 6a) [40], consistent with Fe/2-OG enzymes that hydroxylated the inactivated carbon centers [41]. No conversion from canucin B to canucin A was detected for the incubation of CanE and canucin B. Contrarily, the full-length precursor peptide CanA could be hydroxylated by CanE, demonstrating that CanE carries out hydroxylation on the linear precursor peptide instead of the threaded lasso peptide [40]. In addition, CanE could also hydroxylate the linear core peptide, albeit inefficiently. Further investigation unexpectedly displayed that the addition of CanB1 dramatically enhanced the catalytic efficiency of CanE, indicating that CanB1 acts as a bifunctional protein and facilitates both the proteolysis reaction with CanB2 and the *β*-hydroxylation with CanE (Figure 6a) [40].

These results suggested a new dual path of canucin A biosynthesis. For the main pathway, CanB1 combined with the nascent precursor peptide CanA and then facilitated the tailoring hydroxylation by CanE, following leader peptide liberation by CanB2 and core peptide macrolactam cyclization by CanC. Another minor pathway with lower productivity is that CanE hydroxylated the precursor CanA individually, followed with the recognition of CanB1 and the succedent maturation by CanB2 and CanC. Notably, the possibility of hydroxylating the lasso threaded canucin B to *β-*hydroxylated canucin A is ruled out (Figure 6a) [40]. Considering that the B1 proteins in lasso peptides display comprehensive abilities to facilitate the liberation in various cases [42,43,44,45], B1 proteins from other BGCs including additional tailoring enzymes may perform similar dual functions.

Lasso peptides RES-701s originally isolated from *Streptomyces* sp. RE-896 are regarded as selective endothelin type B receptor (ETBR) antagonists. RES-701-2 and RES-701-4 contain a C-terminal 7-hydroxy-tryptophan compared to the unhydroxylated RES-701-1 and RES-701-3 [46,47,48]. Recently, RES-701-3 and RES-701-4 that differed in the hydroxylation of the C-terminal tryptophan residue were rediscovered through genome mining from the marine *S. caniferus* CA-271066, and their BGC (hereafter termed *res*) was identified with an additional gene (*resE*) encoding a hypothetical protein (Figure 6b). Despite lacking any evidence, ResE was proposed for the 7-hydroxylation of the C-terminal tryptophan residue, which remains to be proved in the future [49].

## 7. Epimerization

The class I lasso peptide MS-271 features the rare non-proteinogenic d-tryptophan at the C-terminus in addition to the two disulfide bonds. The previously identified BGC of MS-271 (*msl*) contained a gene coding for a CapA family protein (MslH) belonging to the metallophosphatase superfamily (Figure 7a), deletion of which completely abolished the production of MS-271 [13]. Homologous genes were also identified in other BGCs such as specialicin [14], aborycin [15], humidimycin [16], and poly-γ-glutamic acid (PGA) [50], a biopolymer that comprises d- and l-glutamic acid connected via amide bonds, but were absent in the BGCs of non-d-Trp containing class I lasso peptides such as sviceucin [17] and arcumycin [18] (Figure 2b).

The function of MslH was further validated for the epimerization of the C-terminal l-Trp in vitro [51]. The full-length precursor MslA is the most favorite substrate for MslH, as the compared reaction with leaderless core peptide only produced a minor amount of d-Trp. Just like CanB1 in canucin A biosynthesis, MslB1 is also a bifunctional protein that not only assists the proteolysis of leader peptide catalyzed by MslB2, but also remarkably enhances the epimerization activity of MslH. Only about 50% conversion of MslA to epi-MslA was observed, implying that MslH generated an equilibrium mixture of the epimers. Since the C-terminal l-Trp derivative has never been detected in the MS-271 producer, the following MslB2 and MslC maturation processes probably recognize epi-MslA as the sole substrate and drive the equilibrium to d-Trp containing precursor peptide (Figure 7b). Furthermore, MslH could epimerize other aromatic residues such as W21F and W21Y at considerable levels, and chimeric substrates with the sviceucin N-terminal core peptide sequence and the C-terminal “CFW” (Figure 2b), displaying a broad substrate tolerance [51].

d-amino acids are limited in RiPPs and only a few mechanisms have been verified. For instance, the single radical *S*-adenosylmethionine (SAM) peptide epimerase PoyD introduces up to 18 d-amino acids in the biosynthesis of polytheonamides [52], another radical SAM epimerase YydG epimerizes the formation of a d-Val and d-*allo*-Ile residues in the biosynthesis of the epipeptide YydF [53]. Additionally, d-Ala and d-amino butyric acid (d-Abu) residues are introduced into lanthipeptides by the hydrogenation of 2,3-didehydroalanine (Dha, dehydrated Ser) and 2,3-didehydrobutyrine (Dhb, dehydrated Thr) via different oxidoreductases, including the zinc-dependent dehydrogenases termed LanJ_A_ [54], the flavin oxidoreductases termed LanJ_B_ [55], and the F_420_H_2_-dependent reductases termed LanJ_C_ [56]. The characterization of the metallophosphatase superfamily protein MslH provides a novel biosynthetic mechanism for d-amino acids in RiPPs.

## 8. Citrullination

Citrullination, referring to Arg deimination to produce non-proteinogenic amino acid citrulline (Cit), had never been reported in RiPPs until the lasso peptide citrulassin A was discovered from *S. albulus* NRRL B-3066 using the Rapid ORF Description and Evaluation Online (RODEO) genome-mining tool. The conversion of Arg9, which is invariable among the citrulassin family, to Cit was certified by in silico analysis of the precursor peptide sequence and nuclear magnetic resonance (NMR) analysis of the maturated citrulassin A (Figure 8b). Heterologous expression of the citrulassin A cluster with ~20 kb upstream and downstream regions only produced *des*-citrulassin A with unmodified Arg9, suggesting the enzyme responsible for citrulline generation is remotely encoded in the genome [22].

Subsequent research revealed that the peptidyl arginine deiminase (PAD) is responsible for deimination of Arg to generate Cit (Figure 8a), as the distantly encoded *pad* gene was ubiquitous in the genomes of citrulassin producing strains with only one exception, while strains lacking *pad* correlated to Arg-bearing *des*-citrulassin production. Heterologous expression of the *pad* gene in native *des-*citrulassin D producer (*S. katrae* NRRL B-16271) resulted in the conversion to deiminated citrulassin D (Figure 8c) [57]. Future work is necessary to unveil the timing of deimination during citrulassin biosynthesis.

## 9. Succinimidation

Protein l-isoaspartyl methyltransferases (PIMTs) usually have a crucial role in protein repair, recognizing and repairing abnormal isoaspartate (isoAsp) residues to l-Asp through a SAM-dependent methyl esterification reaction [58]. In total, 48 lasso peptide BGCs were uncovered bearing genes annotated as *O*-methyltransferases that belong to PIMT homologues, and the extremely conserved Asp6 in all the putative precursor peptides suggested that it might be the modification site [59]. Heterologous expression of two clusters from actinobacterium *Thermobifida cellulosilytica* (*tce*) and firmicute *Lihuaxuella thermophila* (*lih*) (Figure 9a) resulted in the discovery of cellulonodin-2 and lihuanodin, featuring an unconventional succinimide moiety (also known as aspartimide) in the macrolactam ring. It was experimentally proved in vitro that TceM and LihM catalyzed the methylation of Asp6 to the corresponding methyl ester, followed with spontaneous nucleophilic attack of the adjacent Thr7 amino group to form a stable succinimide moiety without further hydrolyzation. Notably, TceM and LihM carried out dehydration on Asp instead of isoAsp, which is in stark contrast to canonical PIMTs. In addition, both TceM and LihM only recognized the threaded lasso peptides rather than linear precursors or isopeptide-bonded rings (Figure 9b) [59].

The functions of TceM and LihM are distinct from the previously reported PIMT OlvS_A_ involved in the biosynthesis of lanthipeptide OlvA (BCS_A_), since the OlvS_A_ catalytic succinimide group was followed with non-enzymatic hydrolysis to either Asp or isoAsp and this process was reversible as isoAsp could be recognized by OlvS_A_ as well to regenerate succinimide [60].

## 10. Linearization

Different from other kinds of tailoring enzymes carrying out various modifications, isopeptidases (IsoPs) are the only post-translational enzymes that take part in the lasso peptide decomposition. The BGCs of astexins from *Asticcacaulis excentricus* feature additional genes *atxE1* and *atxE2* annotated as peptidases, which are homologous to prolyl olipopeptidase (POP) family proteins [61,62]. Incubation of astexin-2 and -3 with AtxE2 showed retention time changes and a mass increase of 18 Da, demonstrating the hydrolyzation of a single amide bond. The identity in both retention times and the MS^2^ spectra with synthetic linear astexin-2 and -3 clarified that AtxE2 is indeed an isopeptidase which linearized the lasso peptides via specific hydrolysis of the isopeptide bonds (Figure 10) [62].

Isopeptidases only hydrolyze lasso peptides derived from the cognate BGCs. As they are located in different lasso peptide BGCs, AtxE1 and AtxE2 showed no hydrolytic activity toward the lasso peptide derived from the other BGC [62]. In total, 24 lasso peptides from different BGCs were used to test the substrate specificity of SpI-isopeptidase (SpI-IsoP) from the BGC of sphingopyxin I (SpI) (Figure 10). None of the substrates except SpI itself could be hydrolyzed by SpI-IsoP [63]. In spite of the narrow substrate specificity, IsoPs exhibited the promiscuous nature as RiPP modification enzymes with substrate mutation variants. The linearized products of loop variants were relatively low, indicating that the loop region may serve as the recognition element for isopeptidase [63,64].

The threaded topology is proved to be necessary for isopeptidase hydrolysis. The hydrolyzation could be detected by retention time changes in HPLC and mass increases in MS^2^ spectra, but no alteration was observed for unthreaded astexin-2 with AtxE2, suggesting the requirement of lariat knot configuration [62]. The crystal structures of AtxE2 and SpI-IsoP showed that isopeptidases consisted of an N-terminal open *β*-propeller domain and a C-terminal *α*/*β*-hydrolase domain [63,65]. The latter featured a conserved Ser-His-Glu/Asp catalytic triad of serine protease, and the isopeptide bond was cleaved via nucleophilic attack by the Ser alkoxide [62,63,65]. Cocrystallization of AtxE2 in complex with tail-truncated astexin-3 further demonstrated that isopeptidase recognizes lasso peptide by shape complementarity rather than specific amino acid sequence, as the Ser10-Gln14 loop region of astexin-3 is suitably accommodated in a narrow and slightly acidic pocket of AtxE2 and a few specific interactions within the complex interface exist [65].

Another intriguing insight into substrate recognition is provided by isopeptidase BenE (Figure 10). Upon heating, benenodin-1 achieved an equilibrium between two threaded conformers with distinct loop region sizes [66]. The conformer 2 more resembles a partially unthreaded peptide wherein the loop is expanded. Particularly, the natural conformer 1 could be cleaved by BenE while conformer 2 within an equilibrium mixture treated under the same conditions could not be processed [66]. It was supposed that the larger loop region of conformer 2 did not fit the active sites of BenE any longer, since the predicted structure of BenE was highly identical to AtxE2 along with the same loop size of benenodin-1 conformer 1 and astexin-3. To further validate the significance of loop size, a chimeric benenodin-1 lasso peptide with the same loop segment of astexin-3 was matured to a threaded structure by benenodin-1 biosynthetic machinery, and then successfully hydrolyzed by both BenE and AtxE2, which had no cross-reactivities toward original substrates, and the benenodin-1/astexin-3 chimera was a more suitable substrate for AtxE2 than BenE [62,66]. Although the ring size of chimeric peptide is one residue smaller than astexin-3, the original substrate of AtxE2, the lasso peptide with correct loop shape could still be recruited into the isopeptidase and effectively cleaved, reinforcing the idea that the loop region of lasso peptides serves as the dominant recognition element for isopeptidase functioning [66].

It seems that the expression and degradation of lasso peptides produced by isopeptidase-containing clusters are under strict regulation. All the isopeptidase clusters uncovered to date share an almost constant gene architecture in which a GntR-like transcriptional regulator is located preceding the precursor gene, TonB-dependent transporter as well as σ/anti-σ factor pairs are found downstream of the isopeptidase gene [61,62,63,66,67,68,69], homologous to the biosynthetic and regulatory system of siderophores to some extent (Figure 10) [62,70]. Phylogenetic analysis exhibited that isopeptidase-containing clusters are a unique clade distinct from BGCs only consisting of class-defining genes or plus ABC-transporter genes. Meanwhile, the distinction between two clades in sequence composition preference and precursor amino acid conservation supported the two-clade model and indicated different evolutional pressures exerted by nature [62]. Notably, the known isopeptidase-containing lasso peptides, except astexin-1 with narrow spectrum activity [61], lack any antimicrobial activity, which are in contrast to the pervasive self-immunity of ABC-transporter-containing lasso peptides [62]. Unexpectedly, the BGC of recently characterized rubrinodin features both an ABC transporter and an isopeptidase, along with a Ton-B-dependent receptor and FecI-/FecR-like regulator pairs, representing an evolutional intermediate of the two-clade model (Figure 10) [71]. It was hypothesized that the hydrolysis of lasso peptides might contribute to the control of intracellular concentration of bioactive lasso products, or isopeptidases may act as factors that combine and release the cargo bound to lasso peptides, related to the roles of siderophores [62], yet the exact biological functionalities of isopeptidase-containing lasso peptides still remain enigmatic.

## 11. Conclusions and Perspectives

Miscellaneous PTMs have always been the highlight of RiPP characteristics, which often bestow diversified generating strategies and improved functionalities on this class of natural products, albeit derived from the limited proteinogenic amino acids. In consideration of their unique lariat knot structure and resistance against thermal and proteolytic inactivation, and especially the eximious physiological functions such as antimicrobial activities against Gram-positive bacteria (e.g., arcumycin [18], LP2006 [22], etc.) and Gram-negative bacteria (e.g., acinetodin, klebsidin [72], citrocin [73], etc.), antitumor activities (e.g., ulleungdin [38], felipeptins [21], etc.), antiviral activities (e.g., MS-271, specialicin [14], etc.) and receptor antagonistic activity (e.g., stlassin [24]), lasso peptides have attracted much attention during the past few years with a notably increasing number of RiPPs of this subclass.

Although no lasso peptides are currently used as marketed drugs or tested in clinical trials as far as we know, it is noteworthy that two antitumor candidates termed LAS-103 and LAS-20x are under investigation by Lassogen, Inc., a company that aim to unlock the potential of lasso peptides for cancer treatment [74]. These compounds further shed light on the promising lasso-peptide-derived therapeutics, as the former is a potent antagonist of ETBR to increase immune response for ETBR-driven malignancies and the latter inhibits the activity of chemokine receptor CCR4 to reduce immunosuppression in the tumor microenvironment [75]. Tailoring PTMs of lasso peptides provides a means to decorate these privileged scaffolds via introducing specific functional groups or altering the structures of the backbone, investigation of which leads to appreciation of the catalytic mechanisms and prospects for bioengineering purposes.

The auxiliary PTMs such as phosphorylation, methylation, hydroxylation, and epimerization recognize the linear precursors ahead of the removal of leader peptides [25,26,31], of which the hydroxylation and epimerization have been confirmed to be promoted by the B1 proteins [40,51]. The possibilities of the promotion by B1 proteins for phosphorylation and methylation cannot be ruled out. We hypothesize that the protein–protein interactions between B1 proteins and the unique post-translational enzymes assist the recognition of precursor peptides. Chances are that the acetylation of albusnodin occurs on the precursor peptide as well, as the absence of *albT* completely abolished the production of albusnodin and no unacetylated intermediate was detected [37]. The succinimidation and linearization, on the contrary, are carried out on the mature lasso rather than linear peptides [59,62,63,66]. There is no clue for the timing of the seemingly spontaneous disulfuration and the non-specific citrullination, and further research is required.

The unusual PTMs mentioned further expand the structural diversity of lasso peptides. The disulfide bridges in class I, III, and IV lasso peptides are of vital importance for the knot-like threaded topology and bioactivities [4,5]. However, the functionalities for other PTMs uncovered up to now, i.e., phosphorylation, methylation, acetylation, hydroxylation, epimerization, citrullination, succinimidation, and linearization, are still to be completely unraveled. It is likely that the test conditions set in vitro do not parallel the biological context in which the PTMs evolved [76]. Since current evidence has affirmed d-amino acids in the stability of peptide structures and resistibility for proteolysis [77], and the antimicrobial activity for Gram-positive bacteria [53,56], the d-Trp residue in some class I lasso peptides might play an analogic role.

The identification of unique PTMs is facilitated by the enzymes generally located in the flanking of BGCs, and the types are relatively predictable through analysis of the protein sequences. The uncovered tailoring enzymes of lasso peptides are limited compared with other highly modified RiPPs, whereas tremendous uncharacterized enzymes have been identified from steadily growing genomic data. The characterized tailoring enzymes can be used as candidates in bioengineering of lasso peptides for optimizing the properties and functionalities. Future study would not only focus on expansion of novel post-translationally modified lasso peptides, but also their catalytic mechanisms, especially for the recognition or interactions among tailoring enzymes, substrates, and other components such as B1 proteins. Moreover, mining peculiar physiological functionalities and better understanding the interactions of tailored lasso peptides with their biological targets would provide more information about the significance of these unusual PTMs.

## Figures and Tables

**Figure 1 ijms-23-07231-f001:**
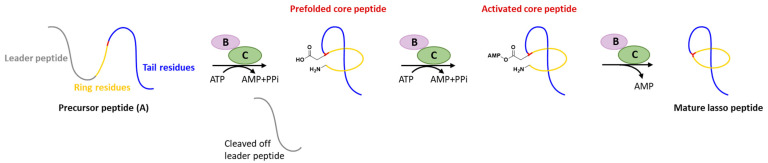
Typical biosynthetic pathway of lasso peptides.

**Figure 2 ijms-23-07231-f002:**
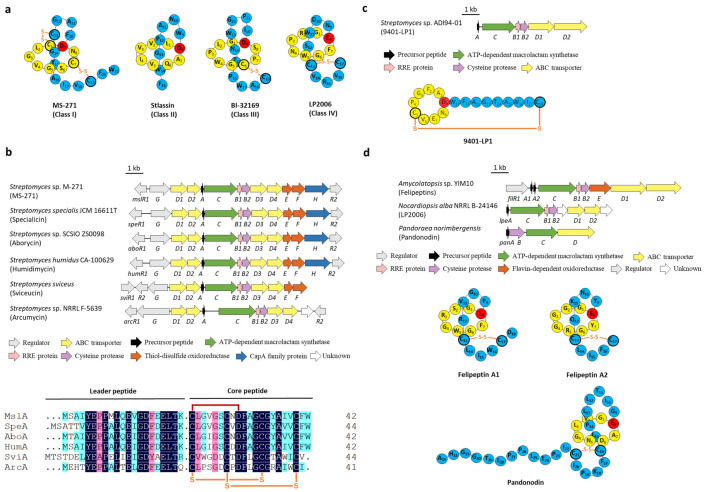
The BGCs and structures of lasso peptides containing disulfide bond(s). (**a**) Representative structures of classes I, II, III, and IV lasso peptides. (**b**) The BGCs of class I lasso peptides and the sequence alignment of the precursor peptides. (**c**) The BGC and structure of class III lasso peptide 9401-LP1. (**d**) The BGCs and structures of class IV lasso peptides.

**Figure 3 ijms-23-07231-f003:**
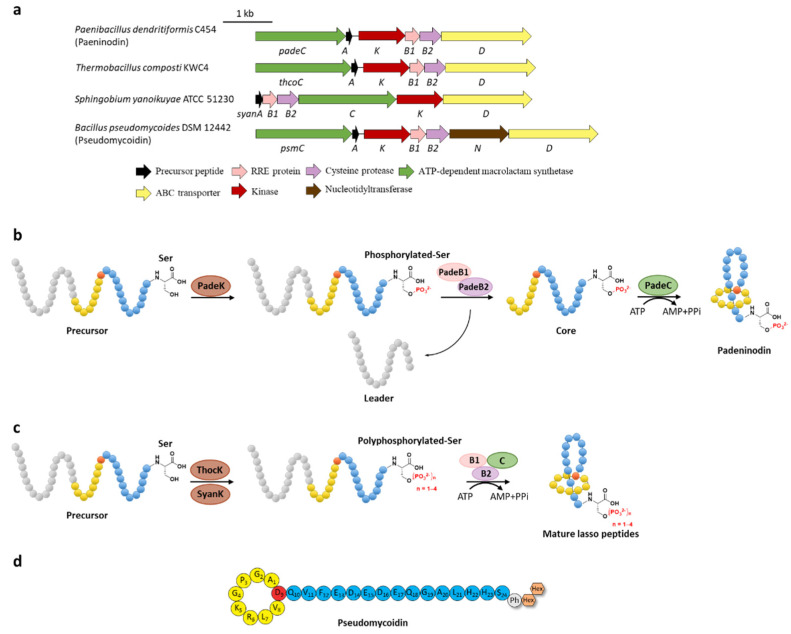
C-terminal phosphorylation of paeninodin and related lasso peptides. (**a**) The BGCs of phosphorylated lasso peptides. (**b**) Proposed biosynthetic pathway of paeninodin. The precursor peptide is phosphorylated by PadeK at the C-terminal Ser residue and then maturated by B1, B2, and C proteins to generate paeninodin. (**c**) Polyphosphorylation of lasso peptides. ThocK and SyanK polyphosphorylated the precursor peptide at the C-terminal Ser residue as well. (**d**) The structure of pseudomycoidin.

**Figure 4 ijms-23-07231-f004:**
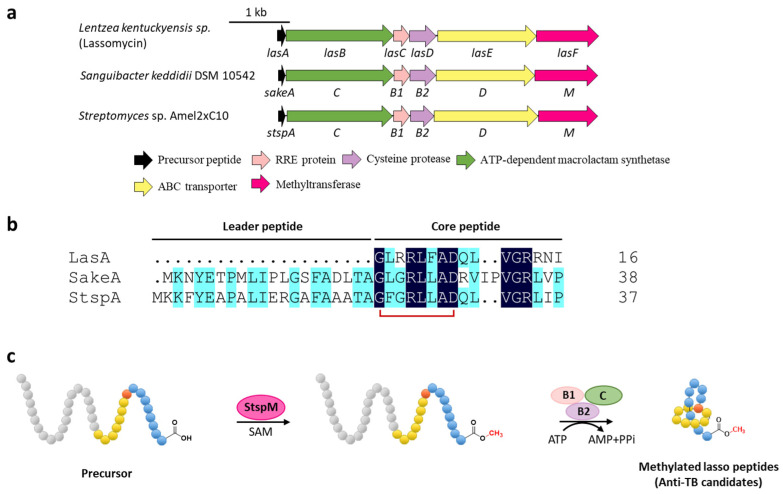
C-terminal methylation of lassomycin and related lasso peptides. (**a**) The BGCs of methylated lasso peptides. (**b**) Sequence alignment of precursor peptides. (**c**) Proposed biosynthetic pathway of C-terminal methylated lasso peptides. StspM methylates the C-terminal carboxyl group of precursor peptide.

**Figure 5 ijms-23-07231-f005:**
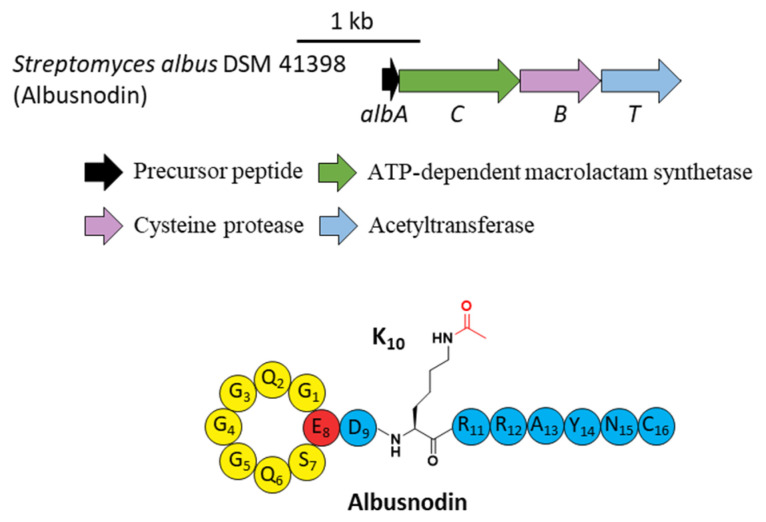
The BGC and structure of albusnodin. The acetyl group attached to the *ε*-amino of K10 is highlighted in red.

**Figure 6 ijms-23-07231-f006:**
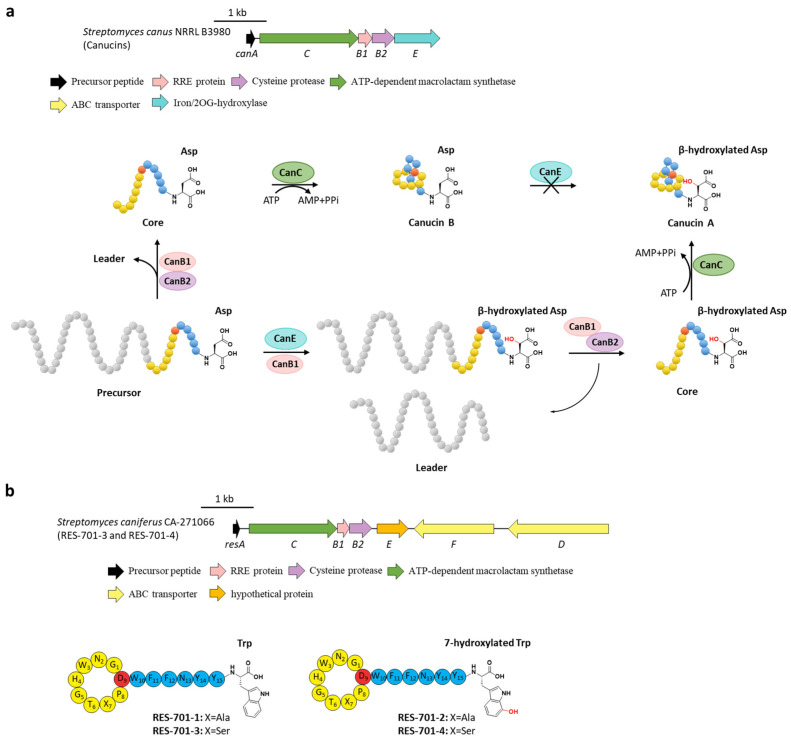
The hydroxylations of canucin A and RES-701s. (**a**) The BGC and distinct biosynthetic pathways of canucin A and B. CanE hydroxylates the *β-*carbon of the C-terminal Asp residue in precursor peptide with the aid of the bifunctional CanB1. (**b**) The BGC and structures of RES-701s, the hydroxyl group in RES-701-2 and RES-701-4 are highlighted in red.

**Figure 7 ijms-23-07231-f007:**
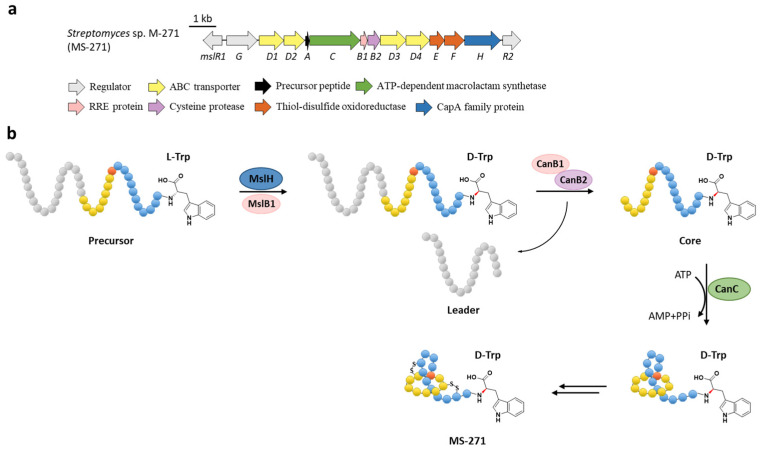
The BGC (**a**) and proposed biosynthetic pathway (**b**) of MS-271. MslH epimerizes the C-terminal Trp residue of precursor peptide with the aid of MslB1, similar to the cooperation of CanE and CanB1 in Section 6.

**Figure 8 ijms-23-07231-f008:**
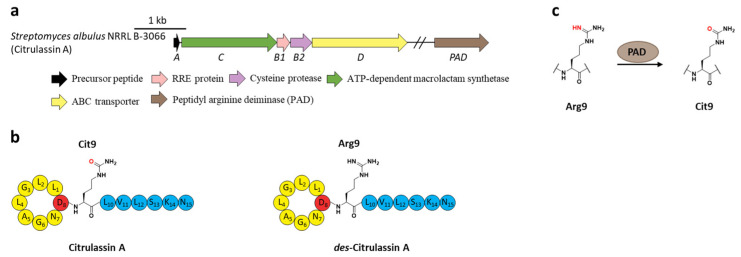
The BGC, structure, and conversion of citrulassin A. (**a**) The BGC of citrulassin A. The *pad* gene is distantly encoded in the genome. (**b**) The structures of citrulassin A and *des*-citrulassin A. The oxygen atom in Cit9 is highlighted in red. (**c**) PAD catalyzes the deimination of Arg9 to generate Cit9.

**Figure 9 ijms-23-07231-f009:**
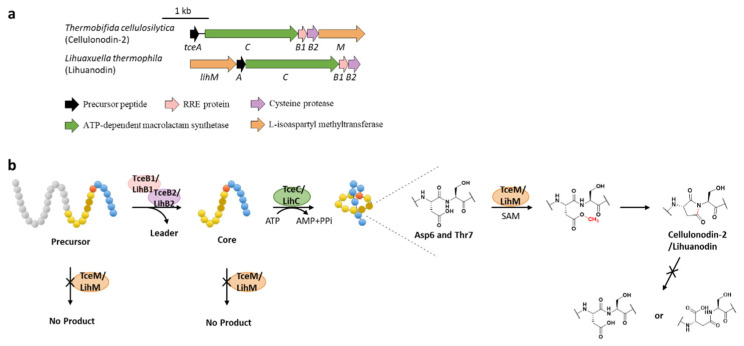
The BGCs (**a**) and biosynthetic pathways (**b**) of cellulonodin-2 and lihuanodin. Zoomed-in images of the Asp6 and Thr7 residues in the threaded lasso peptides are provided for further illustration. Both TceM and LihM could only recognize the threaded lasso peptides as substrates.

**Figure 10 ijms-23-07231-f010:**
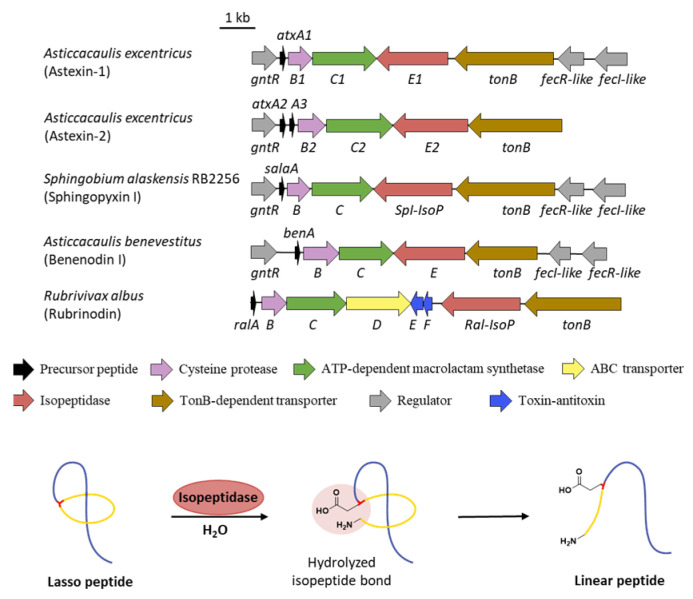
The isopeptidase-containing BGCs and the linearization of lasso peptide. All the isopeptidase-containing BGCs lack the gene encoding ABC transporter except the BGC of the recently characterized rubrinodin. The isopeptidase hydrolyzes the isopeptide bond to reproduce the linear core peptide.

## Data Availability

Not applicable.

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
