# Peer review of "Unusual Post-Translational Modifications in the Biosynthesis of Lasso Peptides"

_ijms, 2022, doi:10.3390/ijms23137231_

Round 1

Reviewer 1 Report

Duan et. al. presented a beautiful summary of lasso peptides which are natural products found in some bacteria. I do not have any major comments except some minor suggestions.

1. I think, the readers would be benefited from having a descriptive figure legends.

2. I just wanted to be clear with the taxonomical acronyms that authors have been using. In some figures 2 a, they used both Streptomyces specialis as well as sp.. Is that for example M-271 is present in multiple Streptomyces species not only specifically in specialis. 

3. Because of their promising therapeutic effects and their stability lasso peptides have been targeted as the attractive candidates for pharmaceutical research. I think, in my opinion, having similar finishing lines within the concluding remarks or somewhere in the current review will also make this review work more significant and timely.

Reviewer 2 Report

1.             General comments 

In this review, the author summarized all the unique PTMs of lasso peptides uncovered to date, with the emphasis on the biosynthetic mechanisms in detail. and the review represents an advance in the understanding the significance of lasso peptides for drug discovery of infectious diseases.

2.   Major revision

1) It is strongly recommended to use the same words both in Fig. 1 and the sentences of lines 35~48, as in the followings.

1-1) Fig.1 and line 36: It is recommended to revise “Precursor peptide” in Fig. 1 to “Precursor peptide (A)”.

1-2) Fig.1 and line 42: It is recommended to revise “the leaderless core peptide” in line 42 to “the leaderless core peptide (Prefolded core peptide in Fig. 1)”.

1-3Fig.1 and line 46: It is recommended to revise “an AMP ester” to “an AMP ester (Activated core peptide in Fig. 1)

1-4) Fig. 1 and line 48: It is recommended to revise “Lasso peptide” in Fig. 1 to “Mature lasso peptide”.

2) It is strongly recommended to show the representative structures of Class I, II, III and IV lasso peptides, so that the reader of IJMS can easily understand the sentences of line 80-88.

3Fig. 4c: It is recommended to explain “D (ABC transporter)” and “Anti-TB drugs” shown in Fig. 4c.
